# Strategies for Sustainable Substitution of Livestock Meat

**DOI:** 10.3390/foods9091227

**Published:** 2020-09-03

**Authors:** Guihun Jiang, Kashif Ameer, Honggyun Kim, Eun-Jung Lee, Karna Ramachandraiah, Geun-Pyo Hong

**Affiliations:** 1School of Public Health, Jilin Medical University, Jilin 132013, China; jiangguihun1@hotmail.com; 2Institute of Food and Nutritional Sciences, PMAS-Arid Agriculture University, Rawalpindi 46300, Pakistan; kashifameer89@gmail.com; 3Department of Food Science & Biotechnology, Sejong University, Seoul 05006, Korea; vollry@sejong.ac.kr (H.K.); eunjunglee@sejong.ac.kr (E.-J.L.)

**Keywords:** livestock, meat consumption, greenhouse gas emissions, climate change

## Abstract

The consequences of climate change are becoming increasingly discernible everywhere, and initiatives have been taken worldwide to mitigate climate change. In agriculture, particularly meat production from the livestock sector is known to contribute to greenhouse gas emissions (GHG) that drive climate change. Thus, to mitigate climate impact, strategies that include a shift in consumption patterns, technological advancements and reduction in food wastes/losses have been discussed. In this review, strategies that focus on meat consumption patterns are evaluated from the technological feasibility, environmental impact and consumer acceptance viewpoints. While plant-based substitutes have efficient nutrient conversion and lower GHG emissions, consumer perception, cost, and other trade-offs exist. Although cultured meat precludes the need of any animals and large land areas, its environmental impact is not clear and is contingent upon production systems and the achievement of decarbonization. Reducing wastes and the re-use of meat processing by-products have the potential to lower the environmental impact. Valuable proteins, heat, electricity and biofuels extracted from wastes and by-products not only reduce the disposal of wastes but also offset some GHG emissions. Perception related challenges that exist for all substitution strategies require specific consumer target marketing strategies. Policy measures such as taxation of meat products and subsidies for alternatives are also met with challenges, thereby requiring reforms or new policies.

## 1. Introduction

The global food production that is dominated by livestock sector is known to contribute to greenhouse gas (GHG) emissions that drive climate change [1]. The impact of climate change (e.g., flooding, droughts, and heat waves) in turn has been shown to negatively affect all dimensions of food security [2,3]. According to an earlier report, livestock production is considered to be a large contributor to climate change accounting up to 14.5% of all anthropogenic GHG emissions [2,4]. Amongst all livestock, beef production is the most problematic due to the requirement of large proportions of resources and land-use as compared to other meat sources [5,6]. Furthermore, the conditions are exacerbated due to water shortages in farming systems and deforestation [7]. However, despite livestock production resulting in significant GHG emissions, climate change mitigation policies have primarily focused on the energy sector and less on the livestock sector [8].

To mitigate climate change, a multitude of strategies have been suggested including development of innovative products, advancement in technologies, reductions in food wastes and losses, and dietary changes [1]. A commonly employed strategy in dietary change is the utilization of plant materials (e.g., soy and peas) for substitution of meat products [9]. It has been highlighted that animal production has higher GHG emission per unit of food output as compared to plant food production [10]. Moreover, from a calorie and land use standpoint, shifting to plant food is beneficial [11]. Other strategies involve materials derived from insects and mycoproteins [9]. Another well-known strategy is cultured meat or in vitro meat, which involves animal muscle cells being formed by tissue culture, so it precludes the need for a whole animal [12]. While the consumption of meat substitutes has grown, the levels of market share of the meat substitutes along with overall consumer utility have been reported hitherto low [13]. Nonetheless, a report has projected that growth in the global protein analogue market will reach $7.5 billion by the end of 2025 [14]. Additionally, it has been suggested that a multifaceted approach that encompasses strategies such as innovative plant- based substitutes, improved waste management and policy reforms, can provide synergistic effects that have the potential to counter the environmental and food security issues [1,15]. Table 1 lists carbon footprints of different meat types along with meat-free products.

In this review, possible strategies for the replacement/reduction of meat are explored. Major strategies that are sustainable and have the potential to mitigate climate change are: (1) plant and non-plant derived substitutes, (2) cultured/in vitro meat production, (3) mini-livestock animals, and (4) utilization of meat processing wastes and by-products. This review primarily focuses on strategies that impact meat consumption patterns from the technological feasibility, environmental impact, and consumer acceptance perspectives (Figure 1).

## 2. Plant-Derived Meat Replacers (Imitation Meat)

A wide variety of plant materials are commonly utilized as meat extenders and replacers (analogues). Meat extenders are supplemented in meat or meat emulsions whereas analogues can be consumed instead of meat products. The key differences are in the resemblance, texture and mouthfeel, which are similar to meat in meat replacers and extenders lacking thereof [18,19]. Meat replacers have been reported with various names in published literature, such as mock meat, imitation meat, meat analogue, faux meat and meat substitutes [20,21].

### 2.1. Technological Feasibility

Some major examples of plant materials that are used to develop meat alternatives include legumes, (lentils, common beans, mung beans), cereals (barley, rye) and oilseeds including rapeseed and cottonseed [18]. However, the concept of meat analogues derived from plant materials is not a new one [22]. Several traditional plant-based meat alternatives are centuries-old and comprise of various types of ingredients [19]. In particular, soy proteins have been commonly used to mimic meat products (animal proteins) owing to their fibrous matrix and gelling properties [23]. Amongst the traditional meat replacers, tofu, which is typically manufactured from soybean curd, is known for its specific bland taste. However, tofu is seen as a meat substitute that is suitable for vegetarians or people with allergies [24]. Tempeh is a popular traditional soybean derived alternative of Indonesian origin. In this fermented product, soybean is used in a whole form as it imparts a dense texture that has the resemblance of meat [25]. Yuba is another popular meat alternative manufactured from soy-based ingredients by accumulating the thin layers of skin that are formed top of soymilk when boiled [26]. The other type of plant-derived meat alternative is Seitan (wheat meat), which is typically manufactured from wheat gluten [19]. Although those mentioned above are more acceptable in some eastern countries, they are largely unacceptable as a meat alternative in several other countries [18]. This had led to the development of meat analogues that mimic the appearance, taste, texture and odor of meat products [27,28].

Textured vegetable protein (TVP) was first recognized in the 1970s and initially perceived as a meat extender. However, TVP had been revived in subsequent years as a meat substitute due to improved product development and marketing strategies [19,21]. Although soy proteins are widely used, other materials, such as wheat, cottonseed and corn have also been attempted [29]. Nonetheless, TVP comprises of processed dried soy flour that imparts a spongy texture and is flavored to improve the sensory properties to resemble meat. TVP can be formed by using a low moisture extrusion process or by fiber spinning [27,29]. Extrusion, which is relatively cost-effective, utilizes soy protein concentrate (protein: ~70%) or soy powder (protein: ~50%), whereas fiber spinning utilizes protein isolates (protein: ~90%) [29]. TVP has been used to formulate products of various shapes (dice, strips, mince and chunks), sizes, colors and textures [30]. However, the use of soy results in the generation of two undesirable off-flavors: grass and bean, (due to the intrinsic lipoxygenases) and bitter and astringent flavor (due to presence of saponins and isoflavones) [18]. Although removal of such unsuitable flavors (compounds) is possible via germination or heat treatment, it is considered undesirable as these phytochemicals possess anticancer activities [31]. Some examples of commonly available plant-based meat substitutes are listed in Table 2.

Another meat substitute that had emerged decades earlier is Quorn. This popular meat substitute mainly contains mycoprotein, which is a hyphae of Fusarium venenatum, particularly from the strain A3/5 (ATCC PTA-2684). Quorn is manufactured with a special emphasis on the development of a texturized product that exhibits maximum resemblance to lean meat in terms of taste and texture [21,32]. Mycoprotein contains high protein and fiber content but low-fat content. This material is produced commercially in bioreactors through continuous fermentation [27].

Even though various types of materials and meat substitutes are available, a widespread substitution has not taken place. This is attributed to various reasons that include lower consumer acceptance and cost-effectiveness. However, pea protein has received increased attention, because it can provide a complementary function to other materials that have unique attributes [22,33]. Another plant material, which is wheat protein, is known to possess the ability to result in fibrous proteinaceous structure due to the formation of disulfide protein linkages [33]. In recent times, some start-ups, such as Beyond Burger and Impossible Foods have successfully formulated plant-based meat products. While Impossible Foods has made use of several plant materials, soy leghemoglobin is particularly used to mimic the characteristic meat color, which is due to myoglobin. In the case of Beyond burger, beet juice extract is used, whereas another venture has used tomato paste [22]. Other examples of materials that have received increased attention include tomato pomace, rice bran and mushrooms. In particular, due to high nutritive and medicinal value, the shiitake mushroom (*Lentinula edodes*) has also been regarded as a promising substitute [34]. In one study, when compared to the German (pork) sausages, edible mushroom (mycelia of *Pleurotus sapidus*) based sausages had better strength and hardness as evaluated by texture profile analysis. Furthermore, the sensory acceptance of mushroom sausages was higher than vegetarian sausages [35]. Another potential alternative is edible seaweeds, which have been utilized as fat replacers in meat products. Studies have shown that incorporation of seaweeds and their derived compounds resulted in organoleptic properties similar to conventional products [36]. Thus, wide varieties of plant materials and methods exist for the development of meat substitutes. Moreover, it is not only more caloric efficient to shift to plant-based products from animal products, but also helpful in the lowering of GHG emissions [37,38].

### 2.2. Environmental Impact

Commercial scale production of plant-based substitutes is mainly accomplished through structuring techniques, such as extrusion, spinning, simple shear flow [39] and bioreactor processing [40]. Extrusion is a thermomechanical process, wherein a combination of heating and mixing plasticizes the proteins placed inside the extruder barrel. The protein melt is then cast into a die resulting in products, such as textured vegetable proteins (TVP) [27]. Although animal derived proteins (e.g., egg, gelatin, collagen, and whey) have been used to mostly electrospin (using a spinneret) fibers, plant derived proteins (e.g., zein and xylan) have also been utilized [41]. Plant materials can also be transformed into fibers by wet spinning [33]. Another method that is used for the commercial production of plant-based substitutes is shear cell technology. Similar to rheometer, a shear cell device consists of a couette cell, wherein the plant material is subjected to a combination of heat and shearing force that results in fibrous structures [33,40]. Bioreactors used for the production of mycoproteins (filamentous fungus) also need additional processes, such as steaming, chilling, and texturization, following fermentation [40]. While the aforementioned techniques can be used to produce plant-based meat analogues on a large-scale (upscaling), they also require large amounts of water along with energy [9,33]. Moreover, the environmental impact of these techniques mainly depends on the energy consumption and specifically on electrical energy, which is related to CO_2_ emissions. Thus, to minimize the environmental impact of extrusion systems, shortening of idle time, (which is when the system is not being used) and reducing the amount of production has been suggested [42]. However, studies on the energy consumption and environmental impact of spinning processes are very scarce. Nonetheless, the environmental impact of a product is quantified by life cycle assessment (LCA), which analyzes a product during its entire life cycle [43].

With regards to plant-based alternatives, Nijdam et al. [44] evaluated life cycle assessment studies (LCAs) of animal and vegetable products. The carbon footprint per kilogram protein of plant-based substitutes was found in the range of 6–17 carbon dioxide equivalent (CO_2_-eq) kg^−1^, and a substitute formed with egg or milk protein ranged from 17–34 CO_2_-eq kg^−1^. The carbon footprint of poultry, pork and beef ranged from 10–30, 20–55 and 45–640 CO_2_-eq kg^−1^, respectively. The lower homogeneity observed with beef than with poultry and pork was attributed to the wide range of production systems for beef [44]. Likewise, in another study, although the replacement of meat with Quorn (mycoprotein) resulted in significant GHG reductions, the GHG footprint of mycoprotein products was similar to poultry. In this study, the GHG reduction was significantly greater for the high social preference scenario (583 Mt CO_2_-eq) as compared to the low acceptability scenario (46 Mt CO_2_-eq) [45]. However to make mycoprotein based protein competitive to chicken, apart from further technological innovation, utilization of agri-food wastes has been recommended. These wastes, as substrates, have been reported to reduce the consumption of energy to 10 kW h, environmental impact to 2–4 kg CO_2_-eq, and land use to 0.5 m^2^ [46]. More importantly, the mycoprotein footprint is mainly from energy consumption, which can be reduced through decarbonization. On the other hand, GHGs from livestock consists of gases other than CO_2_ [47]. However, it is important to note that the environmental impact of the production system depends directly on the type of fossil fuel used in energy generation (electricity). Although the energy consumed by an extrusion system is the same regardless of the country, CO_2_ equivalent emissions vary depending on the country. In the United States and Brazil, the CO_2_ equivalents (kg CO_2_/kg product/day) is about 31,270 and 9130, respectively. The larger CO_2_ equivalents released in the United Stated is attributed to the use of coal for the generation of electricity compared to the use of natural gas in Brazil [48]. However, due to globalization, production and consumption of products occur in different regions of the world.

### 2.3. Consumer Acceptance

Although research on the factors that influence consumers to replace meat with plant-based meat alternatives are limited [13], few studies have shown that plant-based meat substitutes have a lower sensory appreciation and higher prices in comparison with livestock meat [28]. Lower consumer acceptance is also due to lack of familiarity with meat substitutes, perceived lower quality and food neophobia [13]. Furthermore, consumer acceptance of plant-based meat substitutes depends on the recognition that the product be eaten instead of livestock meat [49]. Thus, it is important that the meat analogue be similar to meat in terms of form and shape. Consumers also expect to know how the meat analogue be prepared. In addition, not only appearance, taste and texture of plant-based substitutes but also its meal context (as part of meal and not separately) is crucial in its acceptance [49].

Some studies have indicated that partial meat replacements have proven to be somewhat effective due to the following reasons: (1) reduced quality levels with respect to textural and sensorial attributes and (2) presence of large proportions of meat in recipes and formulations [20]. However, the presence of commercial plant substitutes in the market is relatively recent as compared to meat. Therefore, apart from developing novel meat substitutes, acceptability of plant substitutes depends on projection of a positive image of meat alternatives (low fat, low carbon footprint) [50], better availability of pertinent knowledge about preparation of meals, and decreased pricing index [13]. In addition, reducing the meaning of livestock meat as a status food and elevating awareness about human health issues can contribute to the growth of the substitutes market [24,51].

## 3. In-Vitro/Cultured Meat

Another approach that has the potential to reduce the environmental impact of the livestock sector consists of growing cultured meat [30]. Other reported names for cultured meat include lab-grown meat, synthetic meat, clean meat, cellular meat, and in-vitro or slaughter-free meat [52]. The production of cultured meat involves the cultivation of animal cells in-vitro instead of obtaining them from the slaughtered animals [30].

### 3.1. Technological Feasibility

In 2013, the first patty burger was prepared from lab-grown cells, and since then, a lot of effort has been made to improve the production process [52]. In cultured meat production, few stem cells, which are taken from a limited number of animals, are subjected to multiplication followed by differentiation into myoblasts (muscle cells), which then form myotubes by fusion giving rise to well-grown muscle fibers. After the maturation of these muscle fibers, they are harvested to form a patty to manufacture minced meat resembling a hamburger’s shape. This proliferated multiplication of stem cells and myoblasts provides the possibility of mass production of clean meat without application of growth hormones and with a limited number of animals [53].

According to some studies, the potential benefits of cultured meat include, a significant reduction in GHG emissions, land and water footprint and preclusion of pain and suffering during rearing and killing of a massive number of animals [30,52,54]. It is also reported that culture meat may also enable a considerable reduction in costs incurred during livestock farming and meat production [30]. However, a major demerit of cultured meat has been that it is devoid (or contains low amounts) of fat cells, nerves and blood that is present in conventional meat. To address the lack of adipose tissue, mesenchymal stem cells have been used to culture adipose tissue. Unlike conventional meat, cultured muscle fibers exhibit yellowish color instead of red or pink, which is attributed to suppressed expression of myoglobin due to the presence of ambient oxygen during culture [55,56]. To overcome this hurdle, muscle fibers have been cultured in low oxygen conditions thereby enhancing myoglobin expression and, in turn providing a red or pink color [56]. The other demerit is the lack of tenderization in cultured meat. Muscle tissues of a slaughtered animal contain intramuscular glycogen, which upon conversion to lactic acid lowers the muscular pH followed by enzymatic degradation of muscle fibers. This process directly affects meat tenderness, which is the chief quality trait for consumer acceptability [52].

For the production of culture meat, three main types of systems are available: (1) microcarrier based culture, (2) culture via cell aggregation and (3) bioreactors. Apart from these three systems, new systems are being developed with the goals of scaling up and lowering cost. Nonetheless, in the microcarrier based culture, cells are grown on microcarrier beads (100–200 µm) typically made of polystyrene. As the beads float in a medium agitated by an impeller, gases and nutrients are mixed allowing the growth of cells. Microcarriers provide the benefit of a large surface to volume ratio owing to its spherical shape. In cell aggregation culture, aggregated cells are grown in a medium that is mixed by an impeller. As with the microcarrier system, wherein initial cell concentration affects cell growth, in aggregation culture high initial cell density improves the colonization of cells thereby enhancing cell culture [53]. However, certain challenges, such as cost, apoptotic cells and scalability limitations exist for these two methods [53]. To prevent the apoptosis of cells, Rho-associated protein kinase-inhibitor (ROCKi) is included in the medium [57]. Cells can also be cultured in bioreactors, of which there are many types. An example of a bioreactor is a packed bed type wherein cells attached to microcarriers form the bed of the bioreactor. The growth medium flows in many directions causing a thorough mixing of gases and nutrients. Thus, bioreactor culture owing to high cell densities results in high throughput, making them more suitable for scale-up (Table 3).

Apart from scalability, criteria for the selection of bioreactors must depend on material and labor requirements along with cost. These are known to vary significantly due to a wide range of bioreactors [58].

### 3.2. Environmental Impact

While some studies have indicated the environmental benefits of cultured meat [30,58], others have claimed that although land and inputs needed for cultured meat could be low, energy required may be higher than livestock. In a study by Tuomisto and Teixeira de Mattos [59], the overall environmental impact of in-vitro meat culture was reported to be lower than livestock (conventional) meat production. In this LCA based study of large-scale cultured meat, it was found that production of a 1000 kg cultured meat caused 1900–2240 kg CO_2_-eq GHGs emissions, which was 78–96% lower than conventional meat production in Europe. In this study, cyanobacterial hydrolysate was assumed to be the energy source for cell growth [59,60]. However, it is indicated that the consideration of cyanobacterial hydrolysate or soy hydrolysate as a low-cost nutrition source lacks any strong scientific support [61]. Although, uncertainty originating from many assumptions made in this study was recognized, it was concluded that the environmental impact of in vitro meat was lower than conventional meat [59]. Likewise, in another study by Mattick et al. [60], it was indicated that in vitro meat had a lower global warming potential than beef but not lower than poultry and pork. In this study, several assumptions were also made, which include the presence of the proliferating muscle cells in a bioreactor and glucose being derived from cornstarch and amino acids (peptides) from soy hydrolysates. Contrarily, Smetana et al. [9] reported an adverse environmental impact of lab-grown meat in a LCA based study. In this study, lab-grown meat was compared with other alternatives, such as chicken, gluten, soymeal and mycoprotein-based alternatives. It was found that lab-grown meat usually requires large amounts of non-renewable energy, which poses an environmental issue. The energy was mainly utilized for the cultivation of media for meat growth that accounted for 75% of total consumption of energy from cradle to plate. In addition, the final product frying also led to 6% of cumulative impact. With respect to GHG emissions, lab-grown meat showed the worst performance in comparison with other alternatives [9]. However, lack of clarity with respect to production method and system boundaries assumptions have been noted [12].

It is also important to note that the GHG emissions based on per unit production of cultured meat may vary significantly due to diversity of production systems and employed inputs. It has been argued that the CO_2_ equivalent (CO_2_-eq) based metrics may be misleading because of provision of poor indicators for actual temperature response [62]. Cultured meat production entirely causes CO_2_ emission, which results from energy generation. This is in contrast with beef production that apart from CO_2_ also results in large amounts of CH_4_ and N_2_O. Since methane is not accumulated, cattle production could be less harmful to the environment than cultured meat in the long term. However, climate impact will largely depend on particular production systems and the achievement of decarbonization in energy generation. It is also suggested that a more detailed LCA would provide greater insights regarding cultured meat production and its environmental footprint [12].

### 3.3. Consumer Acceptance

Although cultured meat does not require the slaughtering of animals to derive meat, some negative perceptions have been identified. Studies have shown personal aversions towards lab-grown meat, which is perceived to be unhealthy, unnatural and of low sensory appeal [63]. The lack of consumption of cultured meat among consumers, who were mostly eating vegetarian meals is attributed to its unhealthiness perception. This indicates that vegetarians may not be suitable as a target group for lab-grown meat. On the other hand, meat eaters (conventional meat) may perceive it as a poor alternative to real meat [64]. Moreover, the perception of naturalness, although a fallacy, is also known to influence consumer acceptance. Consumers tend to consider natural food as being always safe and health promoting, whereas synthetic or processed food to be deleterious to health [65]. However, commercialization, improved familiarity, and media coverage could likely influence consumer acceptance of cultured meat [66]. In addition, efforts have been made to lower the cost of lab grown meat as this is considered a key hurdle in its acceptance [63].

## 4. Mini-Livestock (Muscle and Non-Muscle Meat)

Mini-livestock meat is derived from small animals that are bred in captivity. Breeding of animals, such as rodents has been suggested to contribute to food security [67] and reduce hunting of wild animals [68]. Therefore, this section explores mini-livestock animals such as rodents and rabbits as substitutes for poultry. Non-muscle meat such as insects is also explored.

### 4.1. Animal Production and Sustainability

#### 4.1.1. Rodents

Rodents belong to the largest Rodentia order of mammals around the globe, which consists of 2200 species of 480 genera and 30 families. The most prominent animals amongst rodents include: mice and rats, pocket mice; spiny and kangaroo rats, and squirrels, which are capable of thriving in wild regions throughout the world [69]. According to an estimate, rural inhabitants of western Africa get 20%–90% of the total animal protein by consuming rodent species. Rodents exhibit great potential to be employed as an alternative of livestock meat on a commercial scale due to simple growth requirements and high reproduction rates [70]. They also require small spaces, lesser amounts of feed (food wastes), energy and water, as opposed to livestock. A ton of beef and pork requires about 16,000–20,000 m^3^ and 4600–5900 m^3^ of water, respectively [71]. Commonly hunted rodents in Africa include: cane rats/grasscutters (Thryonomys swinderianus and Thryonomys gregorianus) and giant rats (Cricetomys gambianus), which serve as the most important species of bush meat category in terms of trade volume. Cane rat can grow in size up to a total weight of 13 kg and almost 80 million cane rats are captured each year in western Africa, which is equivalent to a meat production of about 300,000 metric tons. Moreover, it is reported that cane rats have high carcass yield (65%), nutritional value and consumer acceptance [72]. In South America, the four most commonly consumed rodents are paca (*Cuniculus* spp.), nutria or coypu (*Myocastor coypus*), agouti (*Dasyprocta* spp.) and capybara (*Hydrochoerus hydrochaeris*). Amongst the rodents, capybara is the largest (35–65 kg) with a high carcass yield, and its meat has been reported to be the most delicious [70,71]. Meat of this rodent is known be lower in cholesterol, leaner and rich in ω3 fatty acids, making it suitable for human consumption [68]. In comparison with large animals, such as ostrich, crocodile, and camel that are marketed to a niche market, rodents offer great potential in commercial production. As a result, strategy to breed rodents has been advocated for several decades [72]. While rodents hold great promise in large-scale commercial production systems, legality issues pertaining to use of some species creates a barrier [73].

#### 4.1.2. Rabbits

The most popular documented family of rabbit and hares is Leporidae, which comprises of more than sixty species and eight different genera found across Asia, Africa, Europe and America [74]. The Californian and New Zealand White breeds have been recognized as leading breeds on a commercial scale [70]. Rabbits are classified as the best meat producers due to rapid adaption, high growth rates, fast production, high water retention and high conversion efficiency (Table 4) [75].

Furthermore, rabbit meat is categorized as white meat along with chicken and turkey, whereas beef, pork, mutton, and horse meat are categorized as red meat [76]. While white meat is perceived as being healthier, a study has indicated that the categorization of rabbit meat into white or red is not straightforward [75]. Aside from being perceived as white, other advantages of rabbit meat include lower cholesterol, lower sodium, high phosphorus content, and higher levels of poly-unsaturated fatty acid (PUFA) and vitamin B [77]. It has been reported that rabbits can convert 20% of consumed proteins into meat, which is comparable with chicken (22%). This is higher than that of pigs (16–18%) and beef (8–12%). Moreover, rabbits are able to utilize proteins derived from cellulose containing plants, while broiler chickens and turkeys require soy-cakes and grains. This utilization of traditional grains put these animals in direct competition with humans for food [82]. Despite these benefits, mass spread of rabbit meat products is impeded by non-competitive costs in comparison with poultry, unavailability of mechanically deboned meat [78] and animal welfare issues [83].

#### 4.1.3. Insects

In some parts of the world (Asia, Africa and Latin America), insects (entomophagy) have been widely appreciated as tasty, wholesome and nutritious sources of high-quality proteins [83]. However, nutrient value varies depending on species, sex, and life stage [80]. Nevertheless, insects are also rich sources of iron, copper, minerals and some vitamins [79]. In recent years, scientists have fostered their efforts to explore insects as food materials and established them as a prominent source of high-quality proteins. According to Alexander et al. [81], proteins from livestock can be substituted with that of insects, such as mealworm larvae and adult crickets, on an equal weight basis. However, the fully domesticated species (bees, silkworms, and cochineals) are only a few [80].

Despite being rich in protein and minerals, a large-scale perception exists that associates consumption of insects to consumption of filth [79,80]. Furthermore, some studies have revealed that apart from the quality of insects, insufficient hygiene, processing and storage influence its consumption, indicating the importance of post-harvest technologies. To this end, a complete change in form has been suggested. It is reported that drying (e.g., sun drying and freeze drying) followed by pulverization of insects into powders or flours (non-recognizable form) could improve their use as food. New processing methods (dry fractionation and ultrasound-based extraction) and disguising insect products (chocolates and ground beef) have been attempted [80]. Other prominent techniques include the introduction of familiar flavors to novel foods that may improve willingness to accept such foods [79]. However, most of the studies have focused on the fortification of foods, such as burgers, bread, pasta and sausages [80]. In a study, edible silkworm pupae up to 15% were added to meat batters for the preparation of frankfurters [84]. In another study, mealworm larvae (90%) were used to produce mincemeat like product. To achieve a meat like aroma, smoking using a wood stove and oil-less frying on hot pan have been attempted [85].

### 4.2. Environmental Impact

A LCA based study by Vayssieres et al. [86] compared the energy use efficiency (EUE) and GHGs of 1 kg rabbit meat with those of beef, dairy, chicken and pork. The total GHG emissions (83.2 kg CO_2_-eq kg^−1^ protein) caused by rabbit meat production was lower than beef production (239.7 kg CO_2_-eq kg^−1^ protein) but higher than pork (35.9 kg CO_2_-eq kg^−1^ protein) and chicken (25.9 kg CO_2_-eq kg^−1^ protein). However, rabbit meat also had a negligible fraction of enteric methane production (2.3%) in comparison with beef (65.5%), dairy milk (26.2%) and pork (6.1%). Furthermore, amongst the animals, rabbits had the highest feed conversion efficiency (FCR) [86], which is considered to be a key factor in the prediction of environmental sustainability and economic success [87]. On the other hand, rabbit meat production had the lowest EUE (0.15 kg crude energy/kg fossil energy), whereas beef had the highest EUE (0.37 kg crude energy/kg fossil energy).

Likewise, in another study by Zucali et al. [88], the global warming potential (GWP) of rabbit meat was compared with those of beef, pork, veal and chicken in Lombardy (Italy). The GWP (kg CO_2_-eq kg^−1^ live weight) of rabbit meat was lower than beef and veal but higher than pork and chicken. Differences in environmental impact are attributed to differences in reproduction, enteric methane production, feed conversion efficiency and rate of mortality. It is important to note that pork and chicken have more homogenous system performances than the production of any other animal [86]. On the other hand, rabbit meat production is still semi-automatic [83]. However, in a recent study, it has been reported that the climate impact of rabbit meat (3.86 kg CO_2_-eq kg^−1^ live weight) production was similar to pork but slightly higher than chicken [87], which has an automatized homogenous system (standardized) [86]. The climate impact of rabbit was found to be dependent on mortality rates and feed substitution. The rate of mortality was highly affected by sanitary conditions, the immune status of animals and age at weaning. Changing feed composition (partially replacing soybean meal with other materials such as sunflower meal, peas, and rapeseed) was shown to lower the environmental impact. Thus, to lower the climate impact of rabbit meat, suitable feeding and management strategies have been recommended [87]. In addition, automatization of rabbit meat production could also result in a lower environmental impact.

Due to the short life cycles, insects are suitable for domestication, which is considered to be less expensive and relatively simple. Insect farming can be undertaken in small spaces and with agricultural wastes. Collection of insects from fields has also been linked to a low environmental impact [80]. Although some studies have recognized insects as an eco-friendly alternative for livestock derived food [89], information on the environmental impact of insect production is very scarce. In one study, using LCA, the environmental impact of production of two tenebrionid species [mealworm (*Tenebrio molitor*) and super worm (*Zophobas morio*)] was compared with chicken, pork and beef. The global warming potential (GWP) of per kg of edible protein derived from mealworm was calculated to be 14 kg of CO_2_-eq, which was found to be lower than that of chicken, pork or beef (data taken from literature). However, energy usage (EU) for mealworm (per kg of edible protein) was similar to pork but higher than chicken and lower than beef. Higher EU was due to the heating needed during growth. To counter this hurdle, inclusion of larger larvae that produce more metabolic heat along with smaller ones has been suggested. Nonetheless, the lower environmental impact associated with insect production was attributed to lack of methane emission, higher reproductive rate and feed conversion efficiency [90].

### 4.3. Consumer Acceptance

Muscle foods, such as rodents and rabbits, and non-muscle foods, such as insects, are all associated with responses of disgust. A major problem in the mass spread of non-traditional meat such as rodent meat is the notion that rats are usually carriers of flees infected with harmful bacteria that may lead to food-borne outbreaks or bubonic plague [71]. However, in areas where rodents are consumed, creating or improving farming systems could lower prices and health concerns. More importantly, farms must undertake biosecurity measures to ensure the safety of animals. Measures must include restricting access to facility/livestock, keeping the facility clean and monitoring animals for signs of infectious diseases [91]. Furthermore, improving processing systems and product presentation could also improve its acceptance among both traditional and non-traditional consumers. Thus, it is suggested that marketing strategies for rodent meat consider animal welfare, cost, quality and propagation of the beneficial uses of meat [68].

Rabbits are considered as pets in many countries, thereby imposing a social barrier to the utilization of rabbit meat [78]. In addition to animal welfare issues, the consumption of rabbit meat has declined in even traditional markets, such as Spain and Italy [92]. Furthermore, non-traditional consumers of rabbit meat are unaware of its nutritive value [83]. To overcome these barriers, it is suggested that the health benefits provided by rabbit meat in history and cultures be incorporated in discourses (story meat) [92]. However, the cost of rabbit meat is relatively higher due to its semi-automatic production. The other impediments to the acceptance of rabbit meat are a perceived wild taste and a general dislike of rabbit meat [83]. Thus, to influence consumer acceptance of rabbit meat, three main strategies have been suggested: intrinsic quality improvement (nutritional improvement through feeding and genetic selection strategies), formation of value-added products (organic farming, product variety) and suitable communication strategies (improved labelling and advertisement campaigns) [83].

Although the nutritive value of insects is recognized, Western cultures regard insects as an unsavory source of food [79]. Even in insect consuming cultures, consumption has declined due to various reasons including westernization and availability of cheaper refined foods [93]. Additionally, issues related to cooking are major hurdles in insect food consumption [79]. Even when powders of edible insects are included in 3D printed food materials, consumer acceptance has been found to be low [94]. The non-adoption and acceptance of insects as livestock meat-replacer foods could be socio-cultural norms, entrenched attitudes at deeper scale and food-supply chain related concerns [79,95]. In a study by Tan et al. [79], it was found that consumer acceptance is affected by three main reasons: (1) issues related to organoleptic and sensory characteristics, (2) safety concerns arising from consumption (physiological harm), and (3) notions about origin and type of insects used as food materials [79]. To overcome the negative perception, increasing exposure to such foods by creating many opportunities to consume them has been suggested [64].

## 5. Exploitation of Meat Processing Waste/Byproducts

Meat industries generate enormous quantities of waste that comprise mainly of organic residues. These types of waste products/by-products do not go through any type of utilization due to several reasons that include lower biological stability, presence of large number of pathogenic microbial loads, high moisture retention, high degradation rates due to enzymatic activities and autoxidation [96,97].

### 5.1. Technological Feasibility

Meat production results in unavoidable wastes that consist mainly of offal (any product other than meat muscles), processing streams (wastewater, exudates, brine solution) [98,99] and packaging materials (paper, plastics, metals, wood). Offal includes edible parts, such as the heart, liver, kidney, intestines, tongue, gizzard and non-edible parts such as hair, bones, skin, and tendons [96]. Based on waste management system, meat-processing wastes similar to other food wastes undergo final processing activities, which include (1) reduce (2) recycle or re-use (3) energy recovery and (4) disposal as shown in Figure 2 [100]. Although it is important to prevent or reduce wastes, management systems should not depend only on these types of strategies. While the reduction of wastes has been shown to likely lower GHG emissions, some amounts of wastes are unavoidable. The re-use of wastes/byproducts as food also faces many hurdles that include consumer perception and regulatory requirements [99,101]. Quality issues and food safety concerns restrict the use of animal-derived by-products in several countries [97,102]. Hence, by-products have found various other uses including pet food, animal feed, and in several other products such as glue, textile, and cosmetics [99]. However, some examples of offal (bovine, ovine, porcine) consumed as food are: liver used in liver sausage and pate, kidney in pies, heart in forcemeat and faggots, tail in soups and stews, blood in sausages, pudding and intestines as sausage casings [99]. Apart from by-products or co-products, meat-processing wastes also contain processing streams, which can also be a valuable source of protein. In one study, proteins recovered from four different co-products and processing streams: blood plasma, exudates, brine solution and stick water, were (instead of porcine meat) incorporated in Irish breakfast type sausages. Proteins derived from processes such as centrifugation, spray drying and freeze drying of co-products and processing streams were able to replace up to 20% of total protein content. Plasma proteins improved essential amino acid content and water-holding capacity while processing stream-derived proteins maintained the overall final quality of sausages at only 10% [103].

### 5.2. Environmental Impact

A study by Winkler and Aschemann [104] employed a Sustainability Impact Assessment approach to evaluate the effect of waste reduction on GHGs emission in Austria. It was estimated that prevention of wastes that is generated during meat processing could save up to 4.8 Mt CO_2_-eq emissions per annum in Austria. This was calculated to be 6% of CO_2_-eq emissions in Austria for the year 2012 [104]. Although, efforts are taken to reduce wastes, meat industry produces large quantities of unavoidable wastes, which are required to be disposed or re-used. With respect to re-use of wastes, in a recent study, proteins were extracted from waste chicken meat via non-thermal pulsed electric fields and mechanical pressing. It was shown that low voltage long pulse treatment was effective in the extraction of liquid from wasted chicken breast muscle without the using any chemicals. About 12% of liquid fraction was extracted from wasted chicken biomass with an investment of total energy of 38.4 ± 1.2 J·g^−1^ [105]. Thus, valuable proteins could be extracted without large investment in energy. Although climate impact was not evaluated in this study, it is likely that any impact could be due to CO_2_ emission originating from energy generation, which could be countered with decarbonization. Nonetheless, it is worthwhile to investigate the climate impact of processes that reduce and re-use meat production wastes/byproducts. Additionally, to avoid large scale dumping of meat processing wastes in landfills, various treatments such as anaerobic digestion, acid hydrolysis, and direct incineration have been employed. These processes not only lower the environmental impact but also provide an energy source [106]. The generation of heat and electricity through animal waste combustion has been incentivized in Europe, because this achieves climate neutrality. According to an estimate, the combustion of 706.5 kg of bone resulted in 155.7 kg of fertilizer and 759.1 kW of heat [107].

### 5.3. Energy Generation

In one study, biogas generated from cattle slaughterhouse waste was utilized to fulfill industrial requirements (self-sufficient). High-fat content (28.4%) found in the slaughterhouse waste streams led to the generation of 641.55 mLCH_4_ g.VS^−1^ of methane through anaerobic digestion [108]. According to an estimate, 1 ton of COD (chemical oxygen demand) subjected to anaerobic digestion can produce up to 350 Nm^3^/h of methane, which can produce 0.15 MW of power. Organic materials such as tallow, blood, and mucosa in wastewater are known to possess high COD and biochemical oxygen demand (BOD) [109]. However, based on net energy analysis, it was shown that energy produced from biogas combustion was able to meet the thermal and electrical demands of the processing facility, thereby helping the industry to be energy self-sufficient [108]. In another study, electricity was generated from poultry manure via fluidized bed combustion. The impact on the environment was noted to be two-fold: (1) savings on fossil fuels, low CO_2_ emission and no wastewater and (2) avoidance of pollutants such as NH3, NOx and N2O as the manure was not spread on the fields [110].

Apart from bio-products, heat, and electricity, biofuels can also be produced from meat processing wastes. Biodiesel, a carbon-neutral fuel that has low sulfur content, CO, and toxicity can be derived from vegetable oils or animal fats. Biodiesel also has been produced from discarded chicken fat, which has cost-effectiveness and ease of availability as a replacer of edible cooking oil [111]. Animal residues contain many types of materials that include water and fat, and it has been estimated that 95% of residual fat can be converted to biodiesel [112]. Biodiesel has also been produced from various sources that include mutton tallow, pigskins and waste fish oil through transesterification [111,113,114]. However, due to the high organic content in waste streams of meat industries (slaughterhouses), anaerobic digestion has been suggested to be more suitable. Materials (heads, condemned meats, spinal cords) that may carry diseases are suggested to be incinerated or dumped in landfills [108].

### 5.4. Consumer Acceptance

Negative consumer perception associated with consumption of by-products is influenced by culture, tradition, religion and ethical constraints [99,101]. Consumption of meat by-products (offal) also faces some hurdles such as unfamiliarity, disgust sensitivity (distastefulness) and ideation. To improve the acceptability of by-products as food ingredients, modifying the physical shape or de-animalizing has been recommended [115]. De-animalization is changing the form of food such as through removal of skin and bones thereby reducing disgust reactions [116]. Nonetheless, a study on the consumer acceptance of offal in South Africa revealed that apart from cost, freshness, and the availability of by-products determined its purchase point (butchery and supermarket). Consumers had a preference for sheep offal over other animal offal [117]. The other major factors that affect the consumption of by-products in humans are nutritional value and regulatory requirements (Table 5).

## 6. Policy Measures

From a policy standpoint, consumer decisions can be influenced by consumer education, financial incentives and regulatory mechanisms. As previously mentioned, policies targeting specific consumer segments, namely: price conscious, healthy eaters, taste driven; green, and organic are suggested to be more beneficial than targeting average consumer. Segment specific education and food labeling regulations have been recommended [13]. However, the meat industries have been reluctant with the adoption of carbon footprint labels [118]. The taxation of meat products not only faces an obvious opposition from the meat industry, but it also affects the low-income consumers. Nonetheless, a holistic approach wherein subsidizing meat substitutes, particularly through personal subsidies (food stamps for low-income segment) along with regulating meat product prices has been suggested for the reduction of conventional meat consumption [13,119]. Studies have indicated that the “meatless days” or “small portions” can be counterproductive. Encouraging consumer to shift to smaller meat portions could lead them to become fixated on meat and even scaring them [20]. Thus, for meat substitution, focus should be on the whole diet and wide variety of consumers. To this end, it has been recommended that the strategies take stakeholders participation, the development of novel meat analogues, and products cost into consideration. More importantly, the success of strategies depends on collective societal effort and a strong push by governmental agencies [20].

## 7. Conclusions

Livestock production for meat is associated with the large use of land, resources, and high GHG footprint along with inefficient nutrient conversion and biodiversity loss. A well-known strategy is the plant-based alternatives, which are mainly derived from soybeans. Cereals, legumes, and mycoprotein also offer novel meat analogues that have the potential to lower GHG emissions and land use along with nutrient conversion efficiency. Although some plant products perform well, low sensory perception, non-competitive cost, and perceived low quality related challenges persist. Another possible strategy involves mini-livestock (animals/insects) that have adapted to marginal or sub-marginal lands. The environmental impact of rabbit meat can be achieved with automatization, improved feeding and farm management strategies. Perceptions of distastefulness, harmfulness, and ideation are major hurdles associated with this strategy. While cultured meat does not require slaughtering any animals, the lack of fat cells, nerves and the tenderization process make it less acceptable. The impact of cultured meat on GHG emissions is not very clear with studies claiming both high and low environmental impacts. Thus, rigorous LCA based studies of cultured meat has been recommended. Due to negative perception, and quality and safety issues, meat by-products have found nonfood uses. Apart from waste prevention, the utilization of waste/byproducts for the extraction of proteins and biofuels have shown to save costs and adapt to climate change. Heat, electricity, and biofuels derived from wastes have the potential to offset the impact of some of the GHG emissions. Consumer perception challenges exist for all the strategies, which are exacerbated by complex interactions between personal preferences and global perspectives. A specific consumer segment targeted marketing strategy has been suggested. Policy measures such as taxation of meat products and subsidies for meat substitutes are met with challenges. Therefore, a multifaceted strategy-based approach is more likely to exert a positive influence on the environmental impact.

## Figures and Tables

**Figure 1 foods-09-01227-f001:**
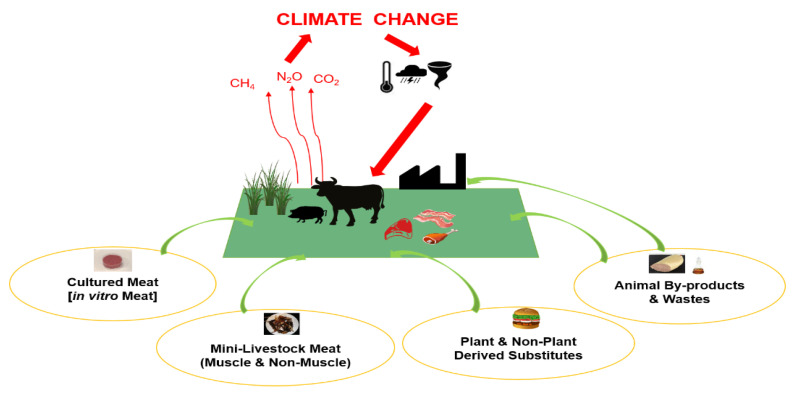
Livestock meat substitution strategies for the mitigation of climate change.

**Figure 2 foods-09-01227-f002:**
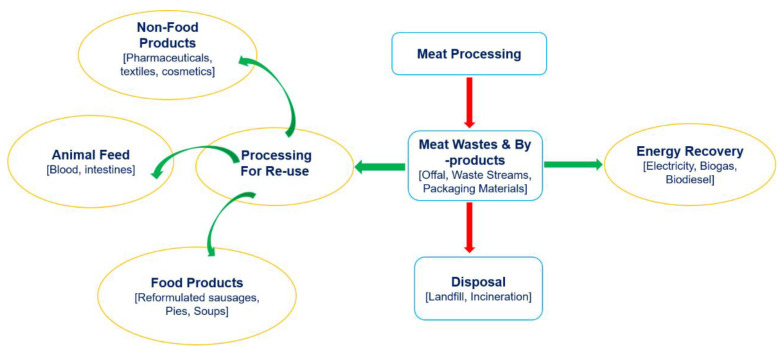
Meat by-products and waste reduction and reutilization strategies.

**Table 1 foods-09-01227-t001:** CO_2_ emissions of major livestock and meat-free products (adapted from Apostolidis and McLeay [13]).

Meat Production Type	kg of CO_2_ per kg of Product	References
Bovine	14–39 kg	[13,16,17]
Ovine (lamb)	39–52 kg	[13,16,17]
Porcine	4–9 kg	[13,17]
Poultry (Turkey)	4–11 kg	[13,17]
Without Meat	2–7 kg	[16,17]

**Table 2 foods-09-01227-t002:** Some examples of commercially available plant-based meat analogues.

Product Name	Compositional Profile	Origin/First Reported Introduction	Energy (Kcal per 100 g)	Protein Content (g/100 g)	Total Fat (g/100 g)	Reference
Quorn^TM^	Mycoproteins derived from fungi.	1st introduced in Europe (1984) and the USA (2002)	141	14.0	2.6	[13,19,32]
Seitan	Comprises of Hydrated wheat gluten	1st use dated to 6th century as an ingredient in Chinese noodles preparation	118	22.1	0.2	[19,22,25]
Wheat Pro ^TM^	Mainly comprised of wheat proteins such as gluten	First reported use was recorded in 1992	408	68	4.0	[19,22,25]
Tempeh	Fermented product made from soy cake	Indonesian product introduced in 1851	180	12	8.3	[19,22,25]
Tofu	Curd derived from soybeans	Dietary staple for >4000 years in china.	120	8.0	4.5	[19,22,25]

**Table 3 foods-09-01227-t003:** Some examples of bioreactors based muscle cell culture (adapted from Moritz et al. [53]).

Bioreactor	Cell Type	Cell Density per mL Medium	References
Packed bed reactor	CHO (Chinese hamster ovary)	2 × 10^7^	[53,58]
Wave bioreactor (Perfusion based)	CHO	2 × 10^8^	[53,58]
Spinner flask	Bovine myoblast	1 × 10^6^	[53,57,58]
CellTank (fixed bed reactor)	CHO	2 × 10^8^	[53,57,58]
Fed batch reactor	CHO	1 × 10^6^	[53,57,58]

**Table 4 foods-09-01227-t004:** Some examples of Mini-livestock animal derived (muscle and non-muscle) meat.

Mini-Livestock Animal	Prospects	Challenges	References
Rodent (*Capybara*)	-High reproduction rates -High carcass yield -High nutritive value -Low cost production -Improved food security	-Safety concerns: dirty meat, carrier of disease -Legal issues with some species -Insufficient knowledge about some species	[67,68,69,73]
Rabbit (*Leporidae*)	-High growth rates -White meat: No adverse health effects -Low cholesterol	-Non-competitive Prices -Lack of deboned meat - Perceived as a pet	[76,77,78]
Insects (e.g., Mealworm larvae and adult crickets)	-High quality protein/nutritive value -Tasty delicacy in some localities -Low cost and space requirements -Short lifecycles	-Negative perception: -Filthy, unsavory food source -Safety concerns (Carriers of diseases and unhygienic meat)	[79,80,81]

**Table 5 foods-09-01227-t005:** Merits and demerits of cultured meat, and meat wastes and by-products.

Meat Substitution Strategy	Prospects	Challenges	References
Cultured meat/ in vitro meat	-Animal rearing Slaughtering of animals not required -Less water and land required -Preclusion of antibiotics, zoonotic disease and fecal matter	-Low quantities/lack of fat, nerves and blood -Lack of meat tenderization -Perceived as unnatural -Energy intensive -Lack of price competitiveness	[30,52,53]
Meat processing wastes and by-products	-Re-use in non-food products (e.g., textile and glue) -Re-use in pet food and pharmaceuticals -Reformulated meat products (e.g., sausages and patties) -Biofuel production -Decreased disposal of wastes	-Safety concerns: pathogens -Lower biological stability -Unappealing sensory properties -Energy usage for extraction of valuable components -Animal suffering not mitigated	[12,96,97,102]

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
