# Peer review of "Strategies for Sustainable Substitution of Livestock Meat"

_foods, 2020, doi:10.3390/foods9091227_

Round 1

Reviewer 1 Report

This is a review of a very actual topic. It is expected that the search for alternatives to meat will have considerable progress in the near future. The article addresses the subject topic accurately and presents results, sometimes divergent, based on recent scientific advances. The different points are discussed on a structure very adequate. Separating rodents and insects at different points is a suggestion for authors to consider. The conclusions are adequate. The tables and figures are relevant but need improvements.

Some detailed comments below:

19 include shift “change with”  include a shift

54 need of a whole “change with” need for a whole

84-85 please rewrite

88 thin layers of skin that is formed “change with” thin layers of skin that are formed

90 Although the aforementioned “change with” Although those as mentioned above,

91 some east countries “change with” some eastern countries,

91 are more acceptable is some “change with” are more acceptable in some

100 cost effective “change with” cost-effective; 118 cost effectiveness “change with” cost-effectiveness

104 off flavors “change with” off-flavors

114 low fat “change with” low-fat

116 wide spread “change with” widespread

122 Beyond burger and Impossible foods “change with” Beyond Burger and Impossible Foods

123 While Impossible foods has made “change with” While Impossible Foods have made

142 Please consider rewriting the sentence. The use of etc. in formal writing is generally frowned upon

142 have also been be “change with” have also been being

149 large amounts “change with” large amounts of

153 system isn’t “change with” system is not

157 Nijdam et al. [41]. Evaluated “change with” Nijdam et al. [41] evaluated

182 plant based “change with” plant-based. Please check the manuscript for similar issue.

196 on projection of positive on “change with” a projection of a positive

218 the considerable “change with” a considerable

Tables captions. Please move to the top of the tables.

252-253 Please rewrite

269 Energy was mainly “change with” The energy was mainly

311 A ton of Beef and pork require “change with” A ton of beef and pork requires

327 Table 2 change with Table 3. Please move the caption to the top of the table.

333-335 check the position of the text.

357 However, the species that have been fully domesticated “change with” However, the fully domesticated species

358-370 please consider rewrite the text to avoid the etc.

374 GHGs of 1 Kg “change with” GHGs of 1 kg

377 negligible fraction “change with” a negligible fraction

386 impact is attributed “change with” impact are attributed

402 an ecofriendly “change with” an eco-friendly

431 higher due its “change with” higher due to its

Please improve the quality of the Figures

462 and 466 Please be consistent in the hyphenation. re-use. Check all manuscript.

468 animal derived “change with” animal-derived

485 Sustainability Impact Assessment (SIA). Please delete SIA. This acronym is used only here. It is not necessary. Check the entire manuscript and change similar issues.

522 cost effectiveness “change with” cost-effectiveness

555 consumer acceptance of offal was observed “change with” consumer acceptance of offal were observed

575 for low income segment “change with” for the low-income segment

605 exert positive “change with” exert a positive

Please check all references and be consistent. For example Ref 59 Annual Review of Earth and Planetary Sciences “change with” Annu

Author Response

Q1: 19 include shift “change with” include a shift

R1: Changed as suggested: Line 19

Q2: 54 need of a whole “change with” need for a whole

R2: Changed as suggested: Line 54

Q3: 84-85 please rewrite

R3: Re-written as suggested: Line 87-88

Q4: 88 thin layers of skin that is formed “change with” thin layers of skin that are formed

R4: Changed as suggested: Line 91

Q5: 90 Although the aforementioned “change with” Although those as mentioned above,

R5: Changed as suggested: Line 93

Q6: 91 some east countries “change with” some eastern countries,

R6: Changed as suggested: Line 94

Q7: 91 are more acceptable is some “change with” are more acceptable in some

R7: Changed as suggested: Line 93

Q8: 100 cost effective “change with” cost-effective; 118 cost effectiveness “change with” cost-effectiveness

R8: Changed as suggested: Line 103 and 121.

Q9: 104 off flavors “change with” off-flavors

R9: Changed as suggested: Line 107

Q10: 114 low fat “change with” low-fat

R10: Changed as suggested: Line 117.

Q11: 116 wide spread “change with” widespread

R11: Changed as suggested: Line 119

Q12: 122 Beyond burger and Impossible foods “change with” Beyond Burger and Impossible Foods

R12: Changed as suggested: Line 125

Q13: 123 While Impossible foods has made “change with” While Impossible Foods have made

R13: Changed as suggested: Line 126.

Q14: 142 Please consider rewriting the sentence. The use of etc. in formal writing is generally frowned upon

R14: Rewritten as suggested: Line 146-148. ‘etc.’ deleted from the manuscript

Q15: 142 have also been be “change with” have also been being

R15: Changed as suggested: Line 148.

Q16: 149 large amounts “change with” large amounts of

R16: Added as suggested: Line 155

Q17: 153 system isn’t “change with” system is not

R17: Changed as suggested: Line 159.

Q18: 157 Nijdam et al. [41]. Evaluated “change with” Nijdam et al. [41] evaluated

R18: Changed as suggested: Line 163.

Q19: 182 plant based “change with” plant-based. Please check the manuscript for similar issue.

R19: Changed here and elsewhere as suggested: Line 187.

Q20: 196 on projection of positive on “change with” a projection of a positive

R20: Changed as suggested: Line 126.

Q21: 218 the considerable “change with” a considerable

R21: Changed as suggested: Line 224.

Q22: Tables captions. Please move to the top of the tables.

R22: Moved as suggested: Line 252.

Q23: 252-253 Please rewrite

R23: Rewritten as suggested: Line 259-261

Q24: 269 Energy was mainly “change with” The energy was mainly

R24: Changed as suggested: Line 276.

Q25: 311 A ton of Beef and pork require “change with” A ton of beef and pork requires

R25: Changed as suggested: Line 318.

Q26: 327 Table 2 change with Table 3. Please move the caption to the top of the table.

R26: Table numbering changed as suggested and captions moved to the top.

Q27: 333-335 check the position of the text.

R27: Text rearranged.

Q28: 357 However, the species that have been fully domesticated “change with” However, the fully domesticated species

R28: Changed as suggested: Line 362

Q29: 358-370 please consider rewrite the text to avoid the etc.

R29: Rewritten as suggested: Line 367-369.

Q30: 374 GHGs of 1 Kg “change with” GHGs of 1 kg

R30: Changed as suggested: Line 379

Q31: 377 negligible fraction “change with” a negligible fraction

R31: Changed as suggested: Line 382

Q32: 386 impact is attributed “change with” impact are attributed

R32: Changed as suggested: Line 391

Q33: 402 an ecofriendly “change with” an eco-friendly

R33: Changed as suggested: Line 407

Q34: 431 higher due its “change with” higher due to its

R34: Changed as suggested: Line 436

Q35: Please improve the quality of the Figures

R35: Figure replaced with new figures: Page: 2 and 13.

Q36: 462 and 466 Please be consistent in the hyphenation. re-use. Check all manuscript.

R36: Consistency checked: all places changed to re-use

Q37: 468 animal derived “change with” animal-derived

R37: Changed as suggested: Line 472

Q38: 485 Sustainability Impact Assessment (SIA). Please delete SIA. This acronym is used only here. It is not necessary. Check the entire manuscript and change similar issues.

R38: Deleted as suggested: Line 488.

Q39: 522 cost effectiveness “change with” cost-effectiveness

R39: Changed as suggested: Line 525

Q40: 555 consumer acceptance of offal was observed “change with” consumer acceptance of offal were observed

R40: Paragraph deleted as suggested by the other reviewer.

Q41: 575 for low income segment “change with” for the low-income segment

R41: Changed as suggested: Line 554.

Q42: 605 exert positive “change with” exert a positive

R42: Changed as suggested: Line 586

Q43: Please check all references and be consistent. For example Ref 59 Annual Review of Earth and Planetary Sciences “change with” Annu

R43: References checked. Abbreviations of journals name added.

Reviewer 2 Report

Comments on the review foods-914032 titled “Strategies for sustainable substitution of livestock meat”

The paper is well written and designed differently from the huge reports that exist in this field. I have only some few comments and suggestions for this review.

For the introduction, the authors can add a table that will summarise the main papers/reports/reviews of the last 10 years that addressed the questions of GHG and meat industry.

Some aspects are missing. I might suggest to the authors to consider “Seaweeds”. Herein an example of report but several studies are available in this field https://doi.org/10.1016/j.tifs.2020.03.039

Further, can the authors consider some statistics for each of the examples “2. Plant-derived meat replacers (imitation meat)”; “3. In-vitro/cultured meat” and “Mini-livestock (muscle & non-muscle meat)”. The statistics might be presented by different manners, around the world or a focus on the top 10 countries in terms of the production or number of studies.

The section ‘6. Marketing Strategies” is not needed and not in accordance to the above. Please remove.

Author Response

Q1: For the introduction, the authors can add a table that will summarise the main papers/reports/reviews of the last 10 years that addressed the questions of GHG and meat industry.

R1: Table 1 included that summarizes GHG emissions of major livestock. References taken from the last 10 years. Table emphasizes types of livestock and its GHG emissions. Table numbers modified as new Table was included.

Q2: Some aspects are missing. I might suggest to the authors to consider “Seaweeds”. Herein an example of report but several studies are available in this field https://doi.org/10.1016/j.tifs.2020.03.039

R2: Brief description about seaweeds and their use in meat analogues added: Line 134-137

Q3: Further, can the authors consider some statistics for each of the examples “2. Plant-derived meat replacers (imitation meat)”; “3. In-vitro/cultured meat” and “Mini-livestock (muscle & non-muscle meat)”. The statistics might be presented by different manners, around the world or a focus on the top 10 countries in terms of the production or number of studies.

R3: Inclusion of Tables was considered as suggested. However, the inclusion of 3 or 4 more tables would necessitate addition of more text to the already long review. Although comparative statistics is very interesting, it could take longer time to create 3/4 tables of Top 10 countries with regards to GHG or consumer acceptance. Since we have time constraints, we have not included the tables.

Q4: The section ‘6. Marketing Strategies” is not needed and not in accordance to the above. Please remove..

R4: The section on Marketing Strategies removed as suggested.